# Invert4TVG: A Temporal Video Grounding Framework with Inversion Tasks Preserving Action Understanding Ability

**Zhaoyu Chen**[1,2,*] **Hongnan Lin**[2,*] **Yongwei Nie**[2,†] **Fei Ma**[1,†] **Xuemiao Xu**[2] **Fei Yu**[1] **Chengjiang Long**[3,‡]

[1]Guangdong Laboratory of Artificial Intelligence and Digital Economy (SZ), Shenzhen, China
[2]School of Computer Science and Engineering, South China University of Technology, Guangzhou, China
[3]ByteDance Inc., San Jose, CA 95110, USA

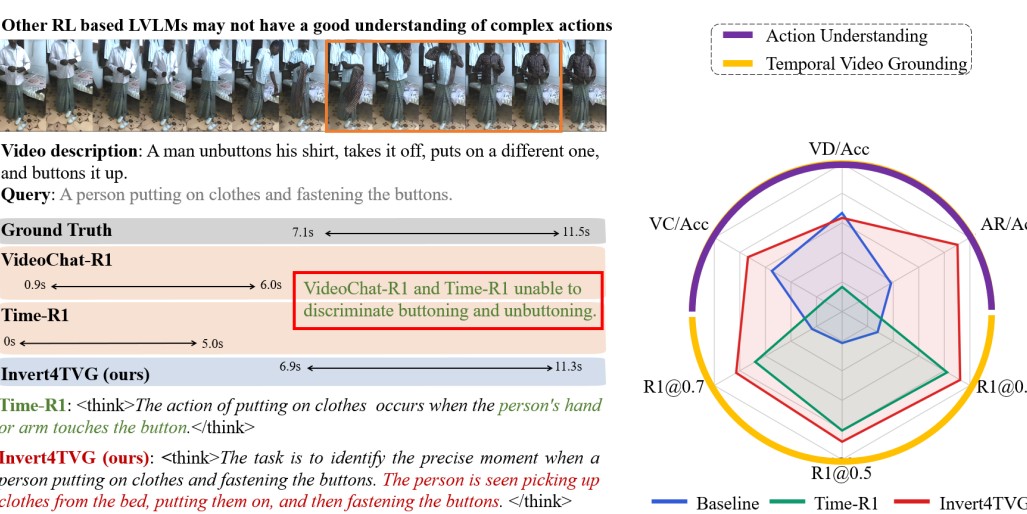

Figure 1: **Left side:** A specific example of temporal video grounding. According to the model's reasoning process,it can be seen that our method achieves better understanding of actions in the video compared to VideoChat-R1 and Time-R1. **Right side:** statistical results demonstrating that Time-R1, which is optimized solely for the IoU loss, reduces action understanding accuracy (where VC, AR, and VD are the proposed three auxiliary inversion TVG tasks measuring multi-granularity action understanding ability). By introducing Inversion-TVG tasks, our method preserves action understanding ability and thus boosts TVG ability (as shown in R1@0.3, R1@0.5, and R1@0.7). Baseline is Qwen-2.5-VL-3B.

## Abstract

Temporal Video Grounding (TVG) aims to localize video segments corresponding to a given textual query, which often describes human actions. However, we observe that current methods, usually optimizing for high temporal Intersection-over-Union (IoU), frequently struggle to accurately recognize or understand the underlying actions in both the video and query, thus reducing the effectiveness of these methods. To address this, we propose a novel TVG framework that integrates inversion-based TVG as auxiliary objectives to maintain the model's action understanding ability. We introduce three kinds of inversion TVG tasks derived from the original TVG annotations: (1) Verb Completion, predicting masked verbs (actions) in queries given video segments; (2) Action Recognition, identifying query-described actions; and (3) Video Description, generating descriptions containing query-relevant actions given video segments. These inversion tasks are

---

*Equal contribution.
†Corresponding authors: nieyongwei@scut.edu.cn, mafei@gml.ac.cn
‡The work was completed when Dr. Chengjiang Long was at Meta.

entirely derived from the original TVG tasks and are probabilistically integrated with them within a reinforcement learning framework. By leveraging carefully designed reward functions, the model preserves its ability to understand actions, thereby improving the accuracy of temporal grounding. Experiments show our method outperforms state-of-the-art approaches, achieving a 7.1% improvement in R1@0.7 on Charades-STA for a 3B model.

# 1 INTRODUCTION

Temporal Video Grounding (TVG) is crucial for long-form video understanding (Gaidon et al., 2013; Laptev & Pérez, 2007; Darrell & Pentland, 1993). It localizes a video segment matching a textual query (Gao et al., 2017; Zhang et al., 2023), enabling applications like video-text retrieval (Zhang et al., 2024) and UAV positioning (Ju et al., 2024).

Existing TVG approaches fall into three paradigms: (1) Traditional methods using hand-crafted features, sliding windows, and DETR-like networks (Shi et al., 2022; Gordeev et al., 2024); (2) LVLMs (Bai et al., 2025; Li et al., 2023) that regress segment duration via pretraining; and (3) RL-finetuned LVLMs, such as Time-R1 (Wang et al., 2025a), using Reinforcement Learning (RL) with format rewards for structured reasoning and IoU rewards for alignment.

Although significant progress has been made, we still find wrong grounding results in existing SOTA methods, and most of these wrong cases stem from incorrect action understanding. Figure 1 left shows one case. In the video, a man unbuttons his shirt, takes it off, puts on another one, and then buttons it up. The query is "A person putting on clothes and fastening the buttons", which requires localizing the actions "putting" and "fastening". Both VideoChat-R1 and Time-R1 notice a hand touching a button and localize the action as "buttoning" rather than "unbuttoning", indicating that they seem to focus only on the button itself without distinguishing between buttoning and unbuttoning. We conjecture that these wrong groundings occur because video grounding models are generally optimized only for IoU. Although IoU is improved, this comes at the cost of reduced action understanding capability, which in turn limits their overall video grounding performance. Figure 1 right demonstrates this statistically. Time-R1 (Wang et al., 2025a), optimized for IoU, shows improvement on TVG metrics (e.g., R1@0.3/R1@0.5/R1@0.7) compared to the baseline Qwen2.5-VL-3B. However, it exhibits degradation in action understanding tasks (VC/VD/AR), which ultimately hampers its TVG accuracy.

A key insight of this work is that training a TVG model effectively requires jointly training auxiliary tasks to preserve the model's action understanding capability. A naive approach to achieving this would be to train the TVG task alongside general action understanding tasks such as action recognition/detection/classification. However, these general tasks are not specifically designed for the temporal video grounding objective, and the understanding learned from them may not align well with the precise temporal localization required in TVG.

Unlike general action understanding tasks, we design action understanding tasks that are specifically tailored for the temporal video grounding task. Specifically, by inverting the input and output of the original TVG task, we convert the localization task into understanding task, obtaining a set of Invert-TVG tasks. A key advantage of these Invert-TVG tasks, compared to general action understanding tasks, is that they share the same training data as the original TVG task. On the same video-query data, our method performs both video localization (via the original TVG task) and action understanding (via the Invert-TVG tasks). This tight coupling enables the learned action understanding to be directly aligned with and supportive of the temporal grounding objective, resulting in more effective and synergistic learning.

Specifically, given a video and a natural language query, the original TVG task predicts the temporal segment duration where the action occurs. Inversely, given a video segment, the proposed Invert-TVG tasks infer the action-related information defined in the query from the given segment. We introduce three Invert-TVG tasks: (1) Verb Completion (VC): mask verbs (actions) in the query and then infer the verbs from video segments. (2) Action Recognition (AR): classify the actions in a given video segment where the ground-truth action is in the query. (3) Video Description (VD): generate descriptions for a given video segment, and the descriptions should contain actions provided in the query.

With the well-defined Invert-TVG tasks, we then propose a reinforcement learning framework that optimizes TVG and Invert-TVG tasks together. However, for large-scale models (e.g., 3B/7B parameters), simultaneously optimizing multiple objectives incurs substantial memory overhead. Moreover, the TVG and Invert-TVG tasks are conflict: the ground-truth video segment that TVG is required to produce is precisely the input to an Invert-TVG task, and the original query that an Invert-TVG task may ask for is exactly the input to the TVG task. To address this, we adopt an alternating optimization strategy, executing TVG and Invert-TVG tasks interleavingly. Besides, since temporal video grounding is our main objective while action understanding is an auxiliary objective, we optimize the TVG task with a higher probability while using a lower probability for the Invert-TVG tasks.

Our contributions include:

- We identify action understanding degradation in TVG from IoU over-optimization, and address the problem via inversion TVG tasks.
- We design three inversion TVG tasks which are self-supervised tasks re-purposing TVG annotations for action understanding, including verb completion, action recognition, and video description.
- We propose a reinforcement learning framework dynamically balancing TVG and Invert-TVG tasks, ensuring robust grounding and understanding.

**Temporal Video Grounding**. Temporal Video Grounding (TVG) (Gao et al., 2017; Hendricks et al., 2017) localizes specific segments in untrimmed videos based on natural language queries. Recent methods fall into two categories: feature-based and frame-based LVLM approaches. Feature-based methods (Carreira & Zisserman, 2017; Lin et al., 2022) extract video and text features using pre-trained encoders, then predict timestamps via multimodal fusion. These rely heavily on feature quality, limiting performance. Frame-based LVLM methods have recently gained traction for their strong generalization capabilities.For instance, NumPro (Wu et al., 2025) introduces a frame-numbering mechanism akin to flipping a manga for efficient temporal grounding, while TimeSuite (Zeng et al., 2024) employs grounded tuning to enhance Large Language Models (LLMs) for long-form video understanding. While methods like these and others (Li et al., 2024; Ren et al., 2024) utilize supervised fine-tuning to generate event sequences, they can still struggle with precise boundary detection on benchmarks like Charades-STA compared to specialized feature-based approaches. To address this, Time-R1 (Wang et al., 2025a) employs reinforcement learning (RL) with IoU rewards, achieving state-of-the-art TVG performance. However, its focus on temporal metrics neglects semantic alignment, constraining long-form video understanding

**RL in LVLMs**. RL has advanced post-training of LVLMs through Reinforcement Learning with Human Feedback (RLHF) (Ouyang et al., 2022; Yu et al., 2024) and Reinforcement Learning with Verifiable Reward (RLVR) (DeepSeek-AI, 2025; Chen et al., 2025). RLHF aligns models with human preferences, improving tasks like image captioning, while RLVR enhances deterministic tasks like visual grounding (Liu et al., 2025). However, RL applications in long-form video tasks remain underexplored due to temporal complexity and semantic challenges. TimeZero (Wang et al., 2025b), Time-R1 (Wang et al., 2025a), VideoChat-R1 (Li et al., 2025) apply RL to TVG but overlooks semantic understanding degradation from IoU-focused rewards. Our Invert4TVG framework addresses this by repurposing TVG data into self-supervised tasks, enhancing action semantic understanding and surpassing traditional RL limitations in video grounding.

## 2 METHOD

The TVG task aims to temporally localize video segments within long-form videos based on natural language queries. Given a video $V$, and a language query $q$, the goal is to identify the temporal boundaries $\tau = [t_s, t_e]$ of the segment of $V$ that best corresponds to $q$, where $t_s, t_e \in \mathbb{R}^+$. The formal definition of the TVG task is as follows:

$$\text{TVG}(V, q) \to \tau. \tag{1}$$

In this work, we introduce Invert4TVG, a framework designed to harness the potential of Large Vision-Language Model (LVLM) for the TVG task using Reinforcement Learning (RL) combined

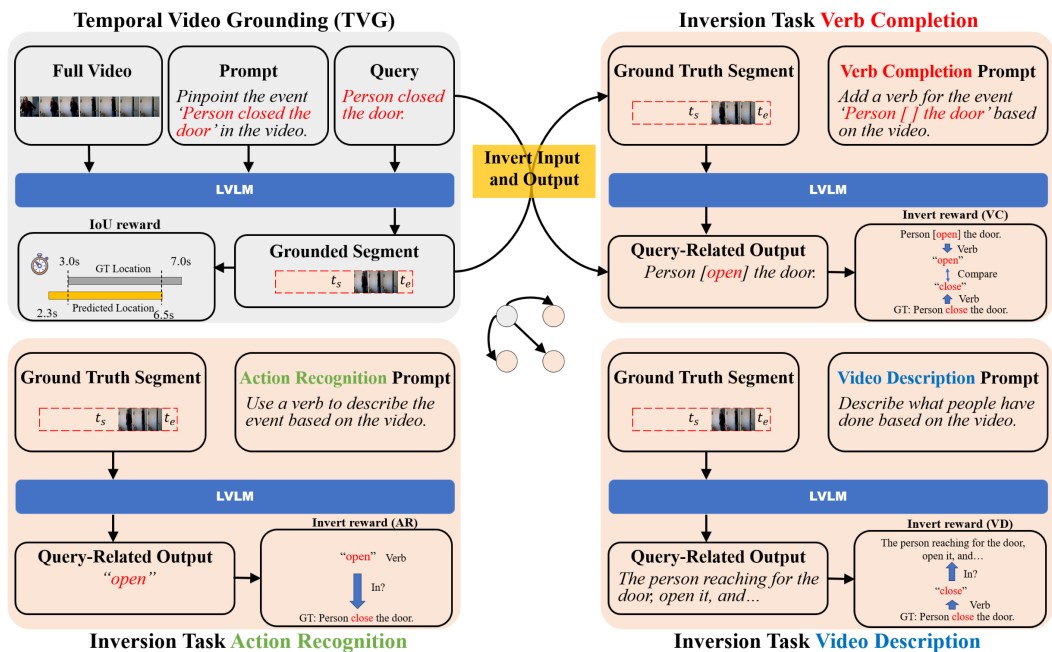

Figure 2: We propose three Invert-TVG tasks. By partially reversing the inputs and outputs of the TVG task we obtain Verb Completion, Action Recognition and Video Description, which reuse the original TVG dataset by taking ground truth video segments as input to reconstruct the target query related actions. The prompts for the three invert-TVG tasks are not identical. For VC, the verb in the query is removed, and the model is required to complete and fill in this verb. AR asks the model to directly estimate the verb in the video. VD requires the model to describe the video content containing action verbs in the query.

with TVG-inversion tasks. The Invert-TVG task is defined as (where $q'$ denotes query-related content):

$$\text{Invert-TVG}(V, \tau) \rightarrow q' \tag{2}$$

Our approach is fundamentally a reinforcement learning algorithm that fine-tunes LVLMs (specifically the Qwen2.5-VL model series) by integrating both TVG and Invert-TVG tasks. In the following, we first introduce the fundamentals of GRPO (i.e., Group Relative Policy Optimization, a reinforcement learning algorithm proposed in (DeepSeek-AI, 2025)). Next, we introduce the proposed inversion TVG tasks, together with reward functions used to train the TVG and Invert-TVG tasks. Finally, we introduce our Invert4TVG reinforcement learning framework.

## 2.1 PRELIMINARY OF GROUP RELATIVE POLICY OPTIMIZATION

DeepSeek-R1 (DeepSeek-AI, 2025), an early R1-style open-source LLM, uses GRPO to train policy $\pi_\theta$ for reasoning before answers. For query $q$, it generates responses $o_1, \ldots, o_G$ with score with $r(\cdot)$, and maximizes:

$$R(o) = \sum_{i=1}^{G} \frac{\pi_\theta(o_i)}{\pi_{\theta_{\text{old}}}(o_i)} \cdot \frac{r(o_i) - \text{mean}\left(\{r(o_i)\}_{i=1}^{G}\right)}{\text{std}\left(\{r(o_i)\}_{i=1}^{G}\right)}, \tag{3}$$

where $\pi_\theta(o_i)$ is generation probability, $\pi_{\theta_{\text{old}}}$ is prior state. The full objective with KL is:

$$\max_{\pi_\theta} \mathbb{E}_{o \sim \pi_{\theta_{\text{old}}}(p)} \left[R(o) - \beta D_{\text{KL}}\left(\pi_\theta \| \pi_{\text{ref}}\right)\right], \tag{4}$$

where $\beta$ is a scaling coefficient. We omit the clipping operation for simplicity.

## 2.2 Invert-TVG Tasks and Reward Functions

In temporal video grounding, the accuracy of model inference highly depends on its understanding of both the video $V$ and the query $q$. Therefore, we do not rely solely on the IoU reward but also introduce additional rewards to keep or even enhance the model's action understanding ability. To this end, as illustrated in Figure 2, we design three Invert-TVG tasks and define reward functions measuring how these tasks are fulfilled. Thanks to advances in NLP, a wealth of mature linguistic toolkits (e.g., SpaCy) can now effortlessly convert verbs into their various tenses or even bring them back to the root form, making it feasible for us to compute reward values for the following inversion tasks.

**Verb Completion (fine granularity).** Verb Completion is a task that masks verbs in the query and asks the model to recover the verbs from the ground truth video segment. For example, if the original query is "Person closed the door", the masked sentence is "Person [ ] the door". The prompt is "Add a verb for the event 'Person [ ] the door' based on the video". As long as the model outputs a sentence successfully recovering the verbs of the ground truth, a reward is given. For instance, if the output sentence is "Person [closes] the door" a full reward is given. Due to the randomness in the model's output, we are primarily concerned with whether the model comprehends the actions within the relevant segments. Therefore, we treat verbs in different tenses as equivalent using SpaCy. The reward function is as follows:

$$
r_{\text{VC}}(o) = \begin{cases} 0 & \text{SpaCy}(v_{\text{pred}}) \neq \text{SpaCy}(v_{\text{gt}}) \\ 1 & \text{SpaCy}(v_{\text{pred}}) = \text{SpaCy}(v_{\text{gt}}) \end{cases}
\tag{5}
$$

where $v_{pred}$ represents the predicted verb in the output sentence $o$, and $v_{gt}$ represents the verb in the ground truth sentence. A full reward is obtained if the root form of the verbs are equal, where $\text{SpaCy}(\cdot)$ brings a verb to its root form.

**Action Recognition (middle granularity).** Action Recognition task trains and preserves the model's action perception capability, and the output of the model is fixed to a single verb. We feed the model the ground truth video segment with prompt as "Use a verb to describe the event based on the video", then compare the predicted verb against any verbs appearing in the ground truth query description. If the model's predicted verb is present in the reference sentence, it receives a full reward. For example, if the model outputs "walk" and the ground-truth sentence is "A person walks away and laughs", a full reward is granted because "walk" occurs in the reference. As before, verb in different tenses are treated as equivalent. The reward function is as follows:

$$
r_{\text{AR}}(o) = \begin{cases} 0 & \text{SpaCy}(v_{\text{pred}}) \notin S_{\text{gt}} \\ 1 & \text{SpaCy}(v_{\text{pred}}) \in S_{\text{gt}} \end{cases}
\tag{6}
$$

where $v_{pred}$ represents the predicted verb $o$, and $S_{gt}$ is the set of root-formed verbs in the ground truth query.

**Video Description (coarse granularity).** Video Description task is employed to train and maintain the model's holistic perception of events, yielding a complete segment level description. Specifically, we feed the model the ground-truth video segment and prompt as "Describe what people have done based on the video". A full reward is granted as long as ground-truth verbs appear in the output sentence of the model. For instance, if the ground-truth verb is "jump" and the model produces "A person jumps and laughs" the reward is awarded because "jump" is present. The reward function is as follows:

$$
r_{\text{VD}}(o) = \begin{cases} 0 & \text{SpaCy}(v_{\text{gt}}) \notin S_{\text{pred}} \\ 1 & \text{SpaCy}(v_{\text{gt}}) \in S_{\text{pred}} \end{cases}
\tag{7}
$$

where $v_{gt}$ represents the ground truth verb, and $S_{\text{pred}}$ denotes the set of root-formed verbs in the sentence $o$ predicted by the model.

## 2.3 IoU and Format Reward Functions

For the TVG task, we mainly employ the IoU reward function. Besides, we introduce a Format reward to enforce the model to output the thinking process.

**IoU Reward**. As stated above, TVG aims at estimating the time interval in the video that is associated with the content of a given textual query. We use the Intersection over Union (IoU) (Yuan

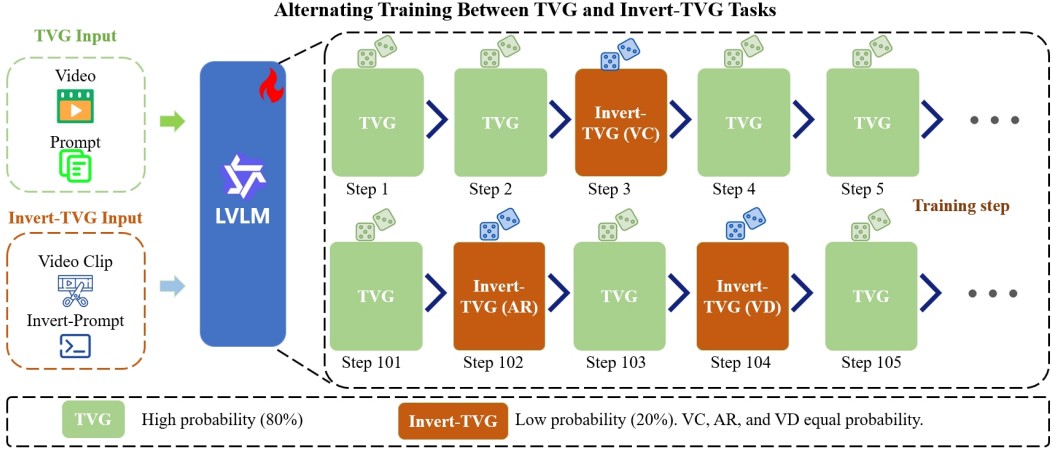

Figure 3: Overview of the proposed Invert4TVG framework. The LVLM dynamically chooses between TVG tasks and Invert-TVG tasks according to different probabilities. Whenever an Invert-TVG task is selected, one of the three variants VC, AR or VD is chosen with equal probability.

et al., 2021) between the time interval predicted by the model and the ground-truth interval as the reward function. This reward function effectively describes the accuracy of the time interval predicted by the model. Given predicted $[t_s, t_e]$ and ground-truth $[t'_s, t'_e]$ segments, the IoU reward can be calculated as follows:

$$r_{\text{IoU}}(o) = \frac{|[t_s, t_e] \cap [t'_s, t'_e]|}{|[t_s, t_e] \cup [t'_s, t'_e]|} \tag{8}$$

where $\cap$ and $\cup$ denote set intersection and union operations.

**Format Reward.** Recently, Time-R1 (Wang et al., 2025a) and VideoChat-R1 (Li et al., 2025) employ a Format reward making the model explicitly output its thinking process before making predictions. Following them, we introduce a template-based reasoning reward that incentivizes the model to generate intermediate reasoning steps prior to providing answers. The format is as following: `<think>***</think> <answer>` $t_s$ to $t_e$ `</answer>`. The reward is formulated as:

$$r_{\text{form}}(o) = \begin{cases} 0, & \text{if } o \text{ has wrong format} \\ 1, & \text{if } o \text{ has correct format} \end{cases} \tag{9}$$

## 2.4 INVERT4TVG REINFORCEMENT LEARNING FRAMEWORK

While a joint training approach that processes all TVG and Invert-TVG tasks simultaneously might seem straightforward, this method suffers from several critical limitations: (1) Memory inefficiency: maintaining separate computation graphs for multiple tasks drastically increases GPU memory consumption; (2) Optimization conflict: gradient updates from different tasks may interfere with each other, especially when their loss landscapes are not aligned; (3) Training instability: the varying convergence rates of different tasks make it challenging to balance their contributions; (4) Task bias: the model may prioritize easier tasks while neglecting others. These drawbacks motivate us to adopt the training paradigm illustrated in Figure 3.

We implement a probabilistic sampling strategy where each training iteration has a high probability (80% in default) of executing the primary TVG task (using IoU and format rewards) and a low probability of performing an Invert-TVG task. When selecting Invert-TVG, we uniformly sample among VC, AR and VD. This design ensures the model maintains its core action understanding capabilities while primarily focusing on temporal grounding. The asymmetric probability distribution prevents the auxiliary tasks from overwhelming the main objective while still providing regular semantic reinforcement. Formally, the reward for training the TVG task is:

$$r_{\text{TVG}}(o) = r_{\text{format}}(o) + r_{\text{IoU}}(o). \tag{10}$$

The reward used to train an Invert-TVG task is:

$$r_{\text{Invert-TVG}}(o) = r_{\text{format}}(o) + r_{\text{inv}}(o), \tag{11}$$

where $r_{\text{inv}}$ is any of $r_{\text{VC}}$, $r_{\text{AR}}$, and $r_{\text{VD}}$. The overall reward function is defined as:

$$r(o) = \alpha r_{\text{TVG}}(o) + \beta r_{\text{Invert-TVG}}(o), \tag{12}$$

where the two coefficients $\alpha$ and $\beta$ take values in $\{0, 1\}$, with the constraint $\alpha + \beta = 1$. Their joint probability distribution is defined as (where $0 \leq p \leq 1$):

$$P(\alpha, \beta) = \begin{cases} p & \text{if } (\alpha, \beta) = (1, 0), \\ 1 - p & \text{if } (\alpha, \beta) = (0, 1), \\ 0 & \text{otherwise.} \end{cases} \tag{13}$$

As mentioned above, $p = 0.8$ is an empirically determined parameter derived from experiments.

## 3 EXPERIMENTS

We now evaluate our Invert4TVG model on the task of temporal video grounding. Code is attached.

### 3.1 EXPERIMENTAL SETUP

**Benchmarks.** We test our model on three temporal video grounding datasets: (1) Charades-STA (Sigurdsson et al., 2016) contains 6,672 long videos capturing indoor human activities. The official split for the TVG task includes 12,408 clip-query pairs for training and 3,720 for testing. (2) ActivityNet (Heilbron et al., 2015) comprises 20K long videos with an average of 3.65 clip-query pairs per video. We use the standard dataset splits with 37,421 training, 17,505 validation, and 17,031 test samples. (3) We further evaluate on QvHighlight (Lei et al., 2021), a high-resolution set of 10,460 long YouTube videos paired with 48k manually annotated clip-queries. To match Charades-STA and ActivityNet formats, multi-segment localizations are split into single-segment tasks, forming a balanced benchmark for fine-grained temporal grounding.

**Implementation Details.** We implement our LVLM using the Qwen2.5-VL model (Bai et al., 2025) as the backbone. To balance efficiency and memory consumption, we sample video frames at 2 FPS and resize them, resulting in approximately 2.8 million pixels per video (e.g., a 50-second video yields 100 frames of size $96 \times 96 \times 3$). Our implementation utilizes SpaCy's en_core_web_sm-3.8.0 model (12MB) to extract verbs from sentences and transform them across different tenses. For optimization, we employ the AdamW optimizer (Loshchilov & Hutter, 2019) with the following parameters: $\beta_1 = 0.9$, $\beta_2 = 0.999$, $\epsilon = 1 \times 10^{-8}$, a weight decay of 0.0, and a learning rate of $5 \times 10^{-5}$. The training time per epoch is approximately 80 hours. To ensure reproducibility, our code, configuration files, and execution scripts are available in the supplementary materials.

**Evaluation metrics.** For TVG, we adopt the "R1@$m$" evaluation protocol to compare with state-of-the-art models, which computes the percentage of samples where the top-1 predicted segment has an IoU greater than a threshold $m$, with $m \in 0.3, 0.5, 0.7$. For brevity, we also adopt mIoU, which calculates the average IoU on all testing data as an alternative metric.

### 3.2 COMPARISON WITH STATE-OF-THE-ART APPROACHES

We compare Invert4TVG with state-of-the-art TVG methods, including both traditional video-language pre-training models (VLP), recent large video-language models fine-tuned via SFT and RL-based approaches.

**Comparisons on the Charades-STA dataset with fine-tuning.** As shown in Table 1, Invert4TVG surpasses not only VLP-based and SFT-based models but also outperforms RL-based approaches under identical conditions. For example, on Charades-STA, the 7B variant of Invert4TVG achieves an R1@0.7 of 51.4, exceeding TimeSuite (43.0), SnAG (46.2), and Time-R1 (50.1). The improvements are more pronounced for the 3B variant. Across R1@0.3, R1@0.5, and R1@0.7, Invert4TVG's 3B model outperforms the 3B version of Time-R1.

**Comparisons on the ActivityNet and QvHighlight datasets in zero-shot settings.** As shown in Figure 4, in zero-shot settings, Invert4TVG's 3B and 7B variants outperform Time-R1 on R1@0.3,

Table 1: Performance of temporal video grounding on Charades-STA. Methods labeled FT with a ✓ were fine-tuned on the Charades-STA training set. Methods marked with * were first pre-trained on extra TVG datasets[1] and then fine-tuned on the Charades-STA training set, while those without * are only trained on Charades-STA. We compare our method against existing 3B, 7B open-source LVLM. We highlight our results and the best-performing baselines using bold and underlining for clear comparison.

| Type | Method | Size | FT | Charades-STA | | |
|---|---|---|---|---|---|---|
| | | | | R1@0.3 | R1@0.5 | R1@0.7 |
| VLP | 2D-TAN* | - | ✓ | 57.3 | 45.8 | 27.9 |
| | Moment-DETR* | - | ✓ | 65.8 | 52.1 | 30.6 |
| | EaTR* | - | ✓ | - | 68.4 | 44.9 |
| | SnAG* | - | ✓ | - | 64.6 | 46.2 |
| SFT | VideoChat-Flash | 7B | | 74.5 | 53.1 | 27.6 |
| | TRACE | 7B | | - | 40.3 | 19.4 |
| | HawkEye* | 7B | ✓ | 72.5 | 58.3 | 28.8 |
| | TimeSuite* | 7B | ✓ | 79.4 | 67.1 | 43.0 |
| RL(3B) | Time-R1(3B) | 3B | | 74.6 | 53.1 | 26.0 |
| | Time-R1*(3B) | 3B | ✓ | 78.7 | 64.1 | 36.9 |
| | Invert4TVG (ours 3B) | 3B | ✓ | **80.8** | **69.0** | **44.0** |
| RL(7B) | Time-R1 (7B) | 7B | | 78.1 | 60.8 | 35.3 |
| | Time-R1*(7B) | 7B | ✓ | 82.8 | 72.2 | 50.1 |
| | Invert4TVG (ours 7B) | 7B | ✓ | **83.0** | **72.5** | **51.4** |

[1] YT-Temporal, DiDeMo, QuerYD, InternVid, HowTo100M datasets, our method is not pretrained on those datasets.

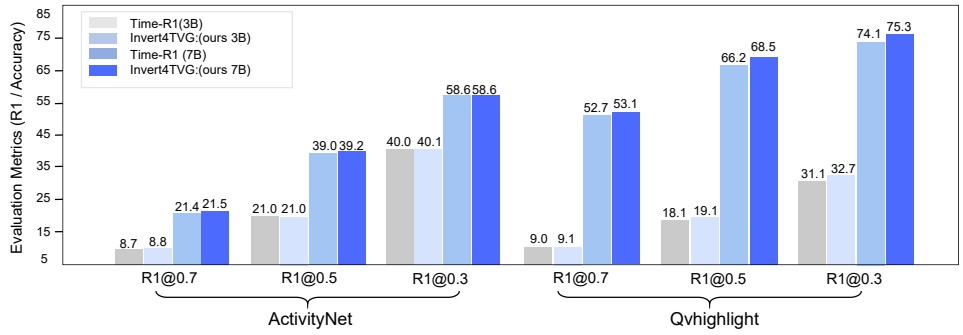

Figure 4: Performance of temporal video grounding on ActivityNet and QvHighlight. We compare our method with Time-R1 (the best-performing among previous methods). All the models are zero-shot tested.

R1@0.5, and R1@0.7 over ActivityNet. On QvHighlight, where we compare single-segment predictions, Invert4TVG consistently outperforms Time-R1 across R1@0.3, R1@0.5, and R1@0.7. ActivityNet contains only 200 action categories, whereas QvHighlight covers a significantly larger and more diverse set, with far more complex scene–action correlations. This disparity underscores the superiority of our method in understanding intricate actions.

## 3.3 ABLATION STUDY

We conduct a detailed ablation on the Invert4TVG-3B model to investigate the contribution of the design strategies.

**Using different combinations of Invert-TVG tasks versus employing them in combination.** As shown in Table 2, using VC, AR, or VD alone instead of jointly yields lower performance. Only-VD, which emphasizes contextual understanding, peaks at R1@0.3 but falls short on precise localization. Only-AR, focused on immediate actions, reaches the highest R1@0.7 of 43.8. Only-VC outputs are less random than Only-VD yet less specific than Only-AR, achieving the best R1@0.5 (68.0). The

mixed-task Invert4TVG surpasses all three individual tasks across all three metrics, demonstrating that joint training outperforms separate use.

The combination of VC and AR improves R1@0.7 to 43.8, outperforming either task alone (VC: 42.0; AR: 43.8), indicating complementary benefits between verb completion and action recognition. The VC+VD pair achieves the highest R1@0.3 (80.0) among two-task setups, suggesting that video description aids verb-focused localization. Invert4TVG (integrating all three auxiliary tasks) achieves the best overall results (R1@0.7: 44.0), demonstrating that multi-task synergy is maximized when all components are jointly optimized.

Table 2: Ablation study using only TVG (Time-R1), VC (verb Completion), AR (action recognition), VD (video description), and their mixed usage.

| Type | Method | Charades-STA | | |
|---|---|---|---|---|
| | | R1@0.3 | R1@0.5 | R1@0.7 |
| RL | Only-TVG | 78.7 | 64.1 | 36.9 |
| | Only-VD | 79.1 | 64.3 | 39.4 |
| | Only-AR | 78.2 | 65.2 | 43.8 |
| | Only-VC | 78.8 | 68.0 | 42.0 |
| | AR+VD | 79.6 | 67.9 | 43.6 |
| | VC+AR | 78.8 | 68.1 | 43.8 |
| | VC+VD | 80.0 | 68.5 | 42.1 |
| | Invert4TVG | 80.8 | 69.0 | 44.0 |

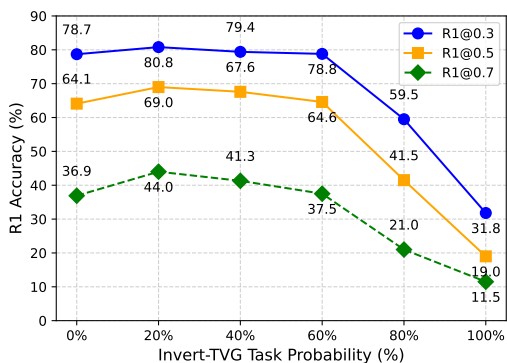

Figure 5: The R1 accuracy curves. Blue, orange, and green show how the three R1 metrics evolve as the Invert-TVG task probability $(1 - p)$ gradually increases.

**Exploiting different probabilities for the TVG and Invert-TVG tasks.** As shown in Figure 5, Varying the task probability markedly alters training outcomes. At 20% Invert-TVG task probability, the model performs best, raising R1@0.7 from 36.9 (no Invert task) to 44.0. As the Invert-TVG task probability grows, the model increasingly emphasizes action recognition while neglecting temporal grounding. Between 60% and 80% Invert-TVG, temporal video grounding performance steadily declines, falling below the pure TVG baseline. When Invert-TVG probability reaches 100 %, the model performs only the Invert-TVG task and yields the worst results. According to the experiments, we choose to set $p = 0.8$ in Eq. 13.

**Binary Invert-TVG rewards vs. cosine similarity-based rewards.** As shown in Table 3, we observe that employing a simple binary Invert-TVG reward (0 or 1) during training yields superior outcomes compared to more intricate reward mechanisms. When training for the same two epochs, the employed Invert-

Table 3: The results using binary Invert-TVG reward or cosine similarity reward for training.

| Reward | R1@0.3 | R1@0.5 | R1@0.7 |
|---|---|---|---|
| Cosine Similarity | 76.2 | 62.2 | 39.8 |
| Binary 0 or 1 | 80.8 | 69.0 | 44.0 |

TVG Reward outperforms the cosine similarity reward across all three evaluation metrics (R1@0.3, R1@0.5, R1@0.7). This advantage stems from the controllability and stability of the binary reward design, whereas cosine similarity introduces higher variance and optimization instability. For ex-

ample, in our implementation, "run" and "eat" yield a cosine similarity of 0.2 despite their weak semantic link. Therefore, binary Invert-TVG reward is a better choice.

## 4   CONCLUSION

In this work, we present Invert4TVG, an approach that introduces Invert-TVG tasks, requiring the model to generate query-related content from a video and its ground-truth temporal segment. We design three variants of Invert-TVG, including verb completion, action recognition, and video description. These tasks encourage the model to retain and enhance its action understanding capabilities. We develop a Invert4TVG RL framework that jointly optimizes TVG and Invert-TVG tasks. In addition to standard IoU and format rewards, we introduce Invert-TVG rewards to promote performance on Invert-TVG tasks. During training, the model primarily performs TVG at a high probability, while intermittently switching to Invert-TVG tasks at a lower probability. This balanced strategy ensures robust temporal localization while preserving semantic action-verb alignment. Our work bridges TVG-LVLM gap, unlocking higher extensions in traditional tasks. Experiments demonstrate the effectiveness of our method over existing approaches, achieving significant improvements of grounding accuracy. The reasoning process also shows that the proposed method indeed understands actions better than compared approaches.

## ACKNOWLEDGMENTS

This research was financially supported by the Open Research Fund from Guangdong Laboratory of Artificial Intelligence and Digital Economy (SZ) under Grant No. GML-KF-24-17, Guangdong Provincial Basic and Applied Basic Research Fund under Grant No. 2025A1515011884, and Fundamental Research Funds for the Central Universities under Grant No. 2024ZYGXZR021.

## ETHICS STATEMENT

This study exclusively utilizes publicly available open-source models and datasets; no proprietary or sensitive information is involved, and all data are free of personally identifiable content. We have strictly followed the corresponding licenses and usage guidelines. Although the present work poses no apparent ethical risks, we caution that—like many machine learning models—its outputs could be misapplied in unforeseen contexts. We therefore advocate responsible use and encourage ongoing efforts to identify and mitigate potential biases inherent in open-source datasets.

## REPRODUCIBILITY STATEMENT

We are committed to ensuring the reproducibility of our work, The models, training datasets, prompts, and hyperparameters used in our experiments are fully documented in Section 4.1 and Appendix C. These descriptions should allow researchers to replicate our experimental setup and-dresults without requiring additional resources beyond those specifed.

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

## A  PRINCIPLE OF ALGORITHM FOR INVERT4TVG

---

**Algorithm 1** GRPO Training with Randomized Invert-TVG Task Selection

---

1: **Input:** Video $V$, query $q$, LVLM parameters $\theta$, probability $p$, reward function $r_{\text{TVG}}(o)$, a list of Invert-TVG tasks $T_{\text{Invert-TVG}} = \{\text{Invert-TVG}_i\}_{i=1}^{N}$ with corresponding rewards $\{r_i\}_{i=1}^{N}$, learning rate $\eta$.
2: **Output:** Optimized LVLM parameters $\theta$ balancing localization accuracy and video-language alignment.
3: Define forward TVG task: $\text{TVG}(V, q) \to \tau$
4: Define a list of Invert-TVG tasks, where each $\text{Invert-TVG}_i(V, \tau) \to q'_i$
5: **while** not converged **do**
6:     Sample a random value $u \sim \text{Uniform}(0, 1)$
7:     **if** $u \leq p$ **then**
8:         Select reward $r = r_{\text{TVG}}(o)$ {Optimize parameters related to $\tau$ (localization accuracy)}
9:         Compute gradient $\nabla_\theta r$ w.r.t. parameters affecting $\tau$
10:     **else**
11:         Randomly sample an Invert-TVG task $\text{Invert-TVG}_i \sim T_{\text{Invert-TVG}}$ {Select a task from the list}
12:         Generate its output $q'_i \leftarrow \text{Invert-TVG}_i(V, \tau)$
13:         Select the corresponding reward $r = r_i(o)$ {Optimize for the sampled Invert-TVG task}
14:         Compute gradient $\nabla_\theta r$ w.r.t. parameters affecting $q'_i$
15:     **end if**
16:     Update parameters: $\theta \leftarrow \theta + \eta \nabla_\theta r$ {Gradient ascent to maximize reward}
17: **end while**
18: **Result:** The model retains both localization accuracy (via $\tau$) and diverse video-language alignment (via various $q'_i$), which are complementary.

---

## B  THEORETICAL JUSTIFICATION FOR MULTI-TASK RL IN INVERT4TVG

To demonstrate the advantages of incorporating the Invert-TVG task into the RL framework, we provide a theoretical analysis showing that the multi-task approach improves semantic alignment and generalization compared to single-task TVG training. We follow the Pareto optimality framework in multi-task reinforcement learning, adapted to our setting where the joint reward balances temporal localization and semantic fidelity.

Let $\pi_\theta$ denote the policy (LVLM), and $D$ the data distribution over videos $V$, queries $q$, and ground-truth segments $\tau$. The single-task objective (TVG-only, as in prior works like Time-R1) maximizes:

$$\max_{\pi_\theta} \mathbb{E}_{o \sim \pi_\theta}[R_{\text{TVG}}(o)] - \beta D_{\text{KL}}(\pi_\theta \| \pi_{\text{ref}}), \tag{14}$$

where $R_{\text{TVG}}(o) = r_{\text{IoU}}(o) + r_{\text{form}}(o)$.

In our multi-task setting, we introduce the joint reward $R_{\text{joint}}(o) = R_{\text{TVG}}(o) + \lambda R_{\text{Invert}}(o)$, with $\lambda > 0$ balancing the tasks. The objective becomes:

$$\max_{\pi_\theta} \mathbb{E}_{o \sim \pi_\theta}[R_{\text{joint}}(o)] - \beta D_{\text{KL}}(\pi_\theta \| \pi_{\text{ref}}). \tag{15}$$

**Lemma 1** (Semantic Alignment Improvement). *The Invert-TVG task minimizes a semantic loss $L_{sem} = \mathbb{E}_{(V,\tau) \sim D}[d(q', q)]$, where $d(\cdot, \cdot)$ is a distance metric (e.g., verb matching or KL divergence on embeddings). Then, the joint loss satisfies $L_{joint} \leq L_{TVG} + C$ for some constant $C > 0$, as $R_{Invert}$ provides positive feedback on semantic fidelity.*

*Proof.* By Jensen's inequality and non-negativity of $R_{\text{Invert}} \geq 0$ (binary rewards in our design), $\mathbb{E}[R_{\text{joint}}] \geq \mathbb{E}[R_{\text{TVG}}] + \lambda \min R_{\text{Invert}} \geq \mathbb{E}[R_{\text{TVG}}]$, assuming $R_{\text{Invert}} \geq 0$. This implies the multi-task policy reduces semantic drift, as Invert rewards enforce alignment (e.g., verb recovery). $\square$

**theorem 1** (Pareto Superiority). *The multi-task policy $\pi_{joint}^*$ is Pareto superior to the single-task policy $\pi_{TVG}^*$ if there exists $\theta$ such that $R_{TVG}(\pi_\theta) \geq R_{TVG}(\pi_{TVG}^*)$ and $R_{Invert}(\pi_\theta) > 0$.*

*Proof.* Consider the convex optimization formulation: minimize $L_{\text{TVG}} + \lambda L_{\text{Invert}}$. Assuming $L_{\text{Invert}}$ is convex (e.g., cross-entropy-like semantic loss), the Pareto frontier dominates the single-task optimum. The KL regularizer ensures the multi-task solution lies on a superior frontier, as feedback from Invert reduces divergence: $D_{\text{KL}}(\pi_{\text{joint}}\|\pi_{\text{ref}}) \leq D_{\text{KL}}(\pi_{\text{TVG}}\|\pi_{\text{ref}}) - \Delta$ for $\Delta > 0$ from semantic regularization. $\qquad\square$

**Corollary 1** (Generalization Bound). *In migrating TVG to LVLMs, semantic drift is reduced by Invert tasks, yielding a generalization error bound: $Err_{joint} \leq Err_{TVG} - \eta\lambda$, where $\eta$ is a learning rate factor derived from multi-task boosting.*

This analysis justifies the inclusion of Invert-TVG, showing improved alignment and generalization on the Pareto front.

## C    MORE IMPLEMENTATION DETAILS

We implement our model using the Qwen2.5-VL model as the backbone, selected for its robust feature extraction capabilities in video understanding tasks. To balance training efficiency and memory constraints, we sample video frames at 2 frames per second (FPS), adaptively resizing each frame to maintain approximately 2.8 million pixels per video. For example, a 50-second video yields 100 frames, each with a resolution of approximately $96 \times 96 \times 3$ pixels. During the reinforcement fine-tuning phase, we train the model for 2 epochs with a batch size of 4. All experiments are conducted on a cluster equipped with eight NVIDIA A100 GPUs (40GB memory each), using CUDA 11.8 and Python 3.10. For natural language processing tasks, we employ the `en_core_web_sm-3.8.0` model from the SpaCy library (12MB) to extract verbs from sentences. Random numbers between 0 and 1 are generated using numpy.random. The model is optimized using the AdamW optimizer with parameters $\beta_1 = 0.9$, $\beta_2 = 0.999$, and a weight decay of 0.0. The learning rate is set to $5 \times 10^{-5}$. Training requires about 80 hours. All code, configurations, and preprocessing scripts are provided in the supplementary materials to ensure reproducibility.

Below, we provide a detailed breakdown of the computational overhead introduced by the three auxiliary tasks (data processing, reward computation, and extra forward/backward passes), including concise cost tables.

| TVG | VD | VC | AR | Total |
|-----|-----|-----|-----|-------|
| 65h | 6.5h | 5h | 3.5h | 80h |

As can be seen from the table, in our method, TVG overall consumed 65 hours (accounting for 81.25%), while Invert-TVG overall consumed 15 hours (accounting for 18.75%), with VD, VC, and AR consuming 6.5 hours, 5 hours, and 3.5 hours, respectively. It is important to note that if the Invert-TVG tasks were not used and only TVG tasks were employed, the total training time would still amount to 80 hours. The Invert-TVG task does not lead to an increase in training time, and our method just took the training time originally meant for the TVG task and used it for the Invert-TVG task.

Detailed time usage in the three auxiliary tasks per training step is shown in the following table:

| Data Processing | Forward | Reward | Backward |
|-----------------|---------|--------|----------|
| VC Offline[*] | 117ms | 81ms | 89ms |
| VD Offline | 121ms | 46ms | 90ms |
| AR Offline | 119ms | 10ms | 90ms |

[*]The data processing was fulfilled before the training.

Similarly, the VRAM costs are nearly identical to the baseline, with no additional overhead.

| Model | VRAM costs |
|-------|------------|
| Qwen-vl-2.5-3B | 146G |
| Our | 142G |

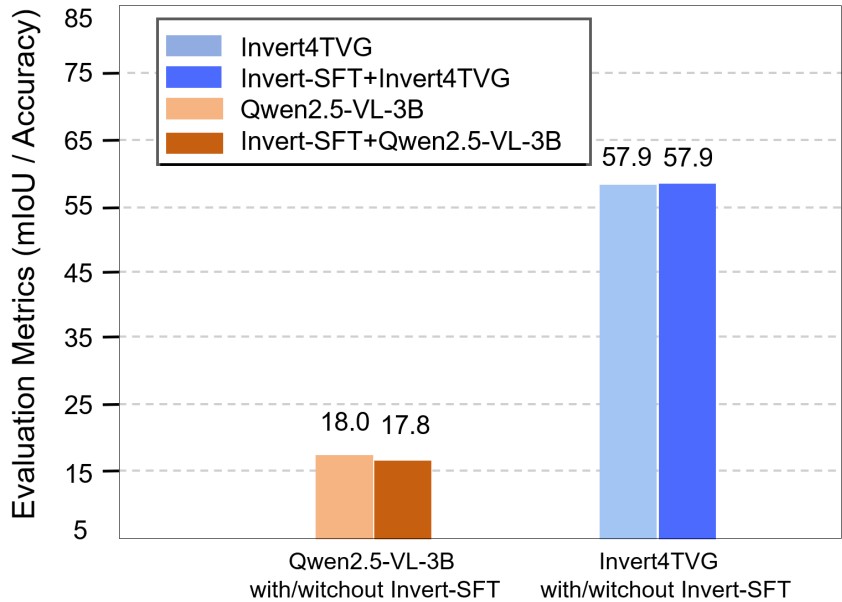

Figure 6: After Invert-SFT, the mIoU of Qwen2.5-VL-3B model and our Invert4TVG method are compared with those without Invert SFT.

## D    ABLATION STUDY

**Impact of Invert-SFT on Model Training.** Invert-SFT refers to feeding ground-truth video clips into the model and requiring the model to output the corresponding event description based on these clips. The ground-truth clips are directly cropped from annotated temporal segments in temporal video grounding datasets, while the "corresponding event" is the query to be localized. After Invert-SFT, the model's outputs become more stable, facilitating subsequent training for Invert tasks. As shown in Figure 6, for the Qwen2.5-VL-3B model initialized with Invert-SFT, the mean Intersection-over-Union (mIoU) slightly decreased from 18.0 to 17.8 initially. However, after sufficient training, both Invert4TVG models, with and without Invert-SFT, reached convergence, achieving identical mIoU scores of 57.9.

## E    QUALITATIVE RESULT

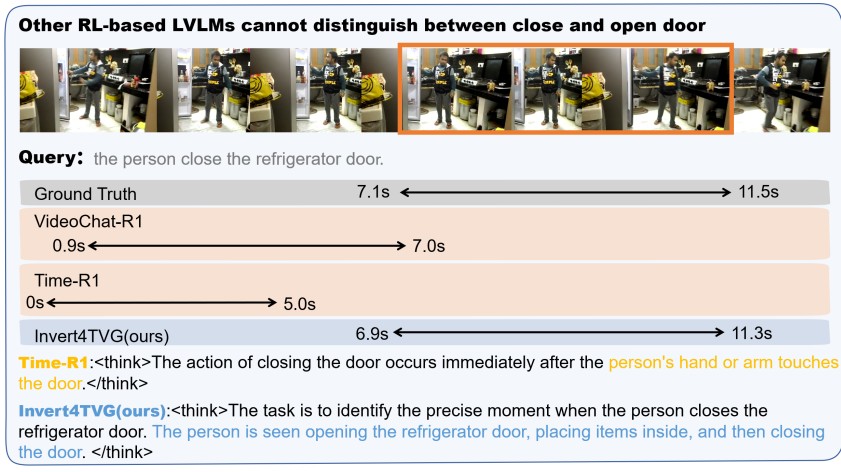

Figure 7: success case 1

**Time localization of similar actions.** As shown in Figure 7, our method can more accurately identify similar actions, such as opening and closing doors. Many other models have insufficient understanding of similar actions, such as picking up and putting things in a box. In the model's judgment, it is likely to be classified as the same action because the model does not fully recognize the state in which the action occurs and continues. Our method can help the model understand actions more deeply and distinguish similar actions.

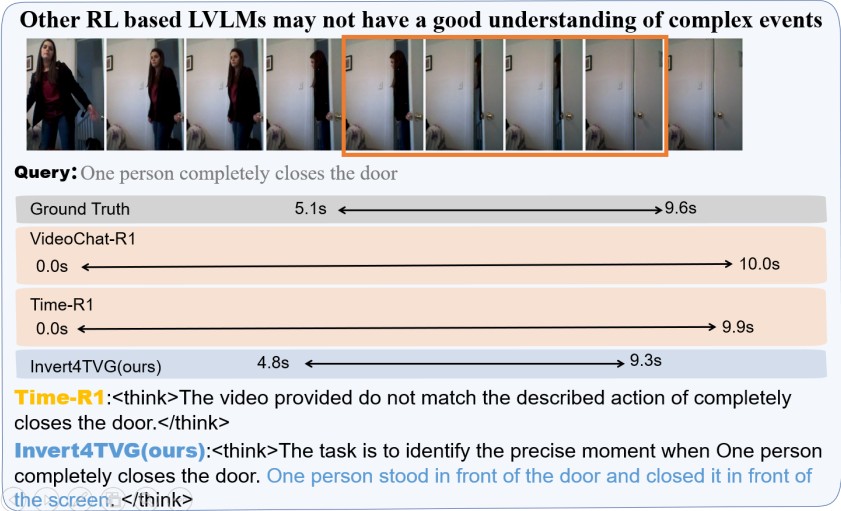

Figure 8: success case 2

**Time localization of complex actions.** As shown in Figure 8, our model recognizes the completion of the door closing action better. For some complex events, the time point we need to locate may occur at the completion of the action, rather than the beginning of the action. Other models sometimes consider the beginning of the action as the time point to be located, and then proceed with subsequent positioning from this time point. Our model has a clearer understanding of the start and end of the action and can effectively locate the time period when the action is in such a state of completion. Meanwhile, a correct understanding of the beginning and end of an action is also helpful for contextual reasoning.

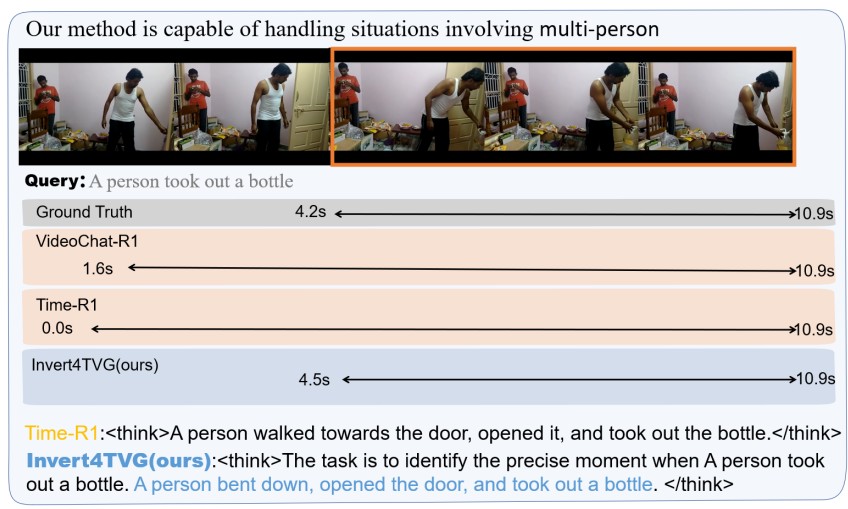

Figure 9: success case 3

**Situations involving multi-person.** As shown in Figure 9, our method demonstrates a strong capability in handling situations involving multiple persons. When two individuals appear in the frame,

other models are often susceptible to interference from the secondary person, leading to extended temporal localization periods. In contrast, our model possesses a more accurate understanding of actions, enabling it to precisely identify the core action to be localized and the state of the target person, thereby achieving superior results.

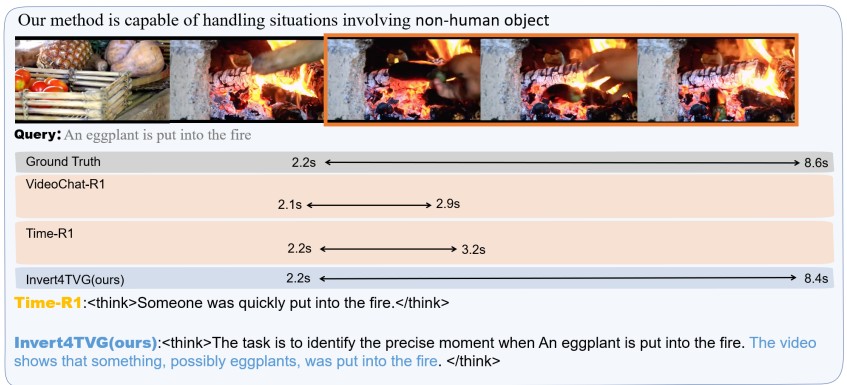

Figure 10: success case 4

**Situations involving non-human object.** As shown in Figure 10, our method is capable of handling scenarios where no person is present in the frame, such as when only a bonfire is shown and an eggplant is thrown into the fire. Other models, upon recognizing the keyword "fire," tend to predict very short temporal segments. Even if these predictions are accurate in timing, their Intersection over Union (IoU) remains low. In contrast, our approach focuses on understanding the action itself, resulting in predicted segments that are longer and closer to the ground truth temporal annotations.

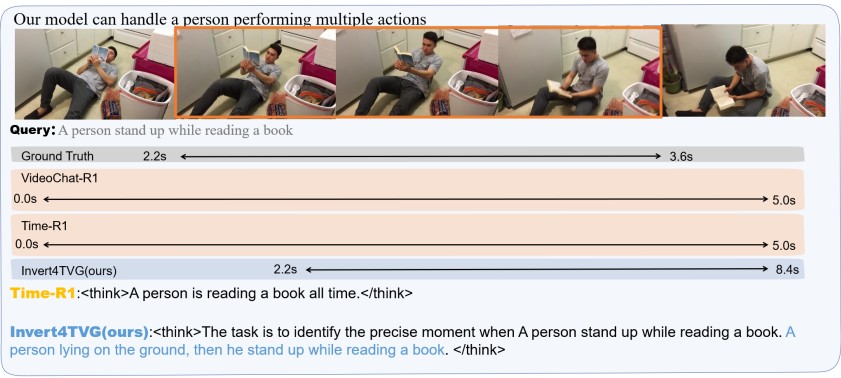

Figure 11: success case 5

**A person performing multiple actions.** As shown in Figure 11, the event to be localized involves a person reading a book while standing up. Other models focus only on a single action, namely "reading," whereas our model disambiguates the actions, accurately identifying both the reading and the simultaneous act of standing up.

**A person performing multiple actions.** As shown in Figure 12, the query contains a temporal cue such as "and then." While other models treat the consecutive actions as a single event and attend more to the earlier action, our model recognizes both actions and their temporal order, yielding a more accurate localization.

**A person performing multiple actions.** As shown in Figure 13, the event to be localized involves a causal relationship: a person in the video first sneezes and then takes medicine. Other models fail to accurately recognize the action of sneezing, leading them to rely on speculation and only localize the action of taking medicine. In contrast, our model successfully identifies both sneezing and taking medicine, understands the causal relationship between them, and achieves superior temporal localization results.

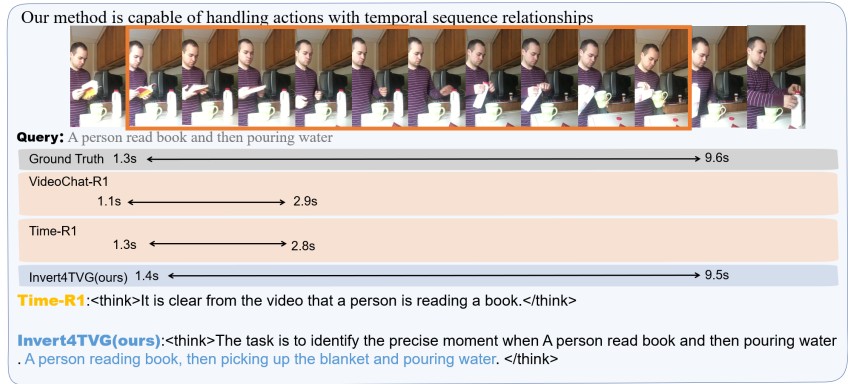

Figure 12: success case 6

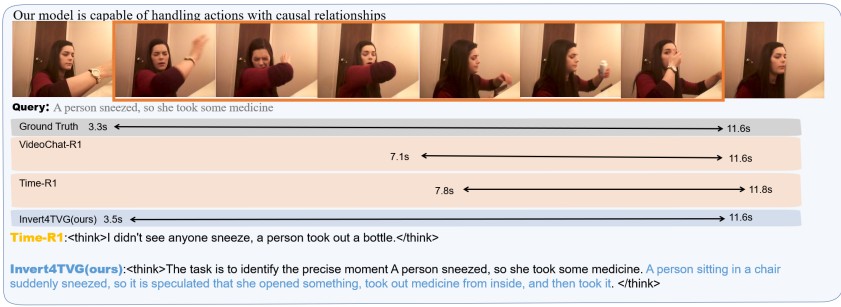

Figure 13: success case 7

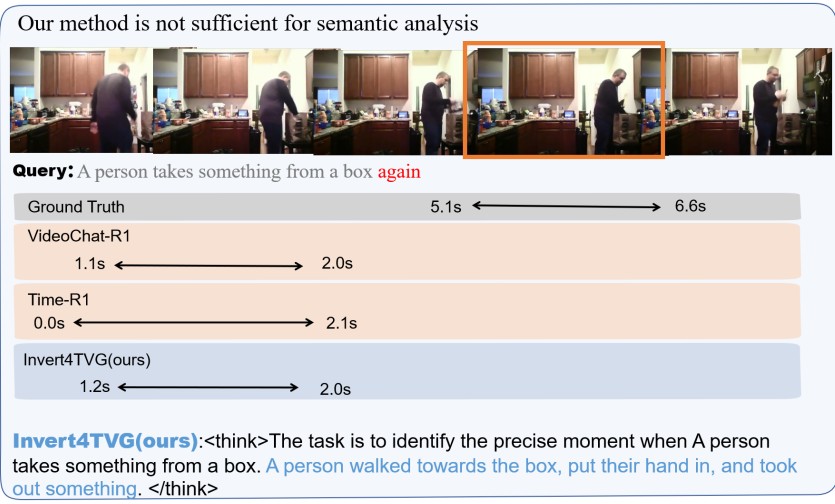

Figure 14: failure case

**Deep understanding of event semantics.** As shown in Figure 14, our method is not sensitive enough to some qualifiers, such as "again" representing the second occurrence of an action, which requires the model to accurately identify the action while also accurately finding the time period during which the second action occurred. Our method, as well as other models, has some shortcomings in this aspect. When locating time, we may find the first occurrence time as the final answer. This is because the model does not have a deep understanding of the meaning of qualifiers in the query and fully considers it when locating video time.

## F    USE OF LARGE LANGUAGE MODELS

We used large language models (LLMs) in a limited and auxiliary manner during the preparation of this paper. Specifically, LLMs were employed to improve the fluency and readability of the manuscript by polishing grammar and style, without altering the technical content. Importantly, LLMs were not involved in formulating research ideas, designing methods, conducting experiments, analyzing results, or drawing conclusions. All technical contributions of this paper are solely the work of the authors.

