# OpenReview forum: "Invert4TVG: A Temporal Video Grounding Framework with Inversion Tasks Preserving Action Understanding Ability"
_ICLR.cc/2026/Conference — ICLR 2026 Poster_

### Official Review · Reviewer_9Nkj · 2025-10-27

**Soundness:** 3
**Presentation:** 3
**Contribution:** 3
**Rating:** 6
**Confidence:** 4

**Summary:**

The paper presents Invert4TVG, a novel RL-based framework for Temporal Video Grounding that introduces three inversion tasks: Verb Completion (VC), Action Recognition (AR), and Video Description (VD). These auxiliary tasks aim to preserve and enhance the model’s action understanding ability, which is often degraded when optimizing only for IoU-based rewards. The method alternates between TVG and Invert-TVG tasks with probabilistic scheduling and achieves consistent performance improvements across datasets such as Charades-STA, ActivityNet, and QvHighlight.

**Strengths:**

1. Strong conceptual motivation. The paper convincingly argues that existing TVG models fail primarily due to a lack of action understanding, not just poor temporal localization. The inversion-based idea to restore semantic understanding is clear and innovative.
2. Clear RL framework. The Invert4TVG reinforcement learning setup is well structured and explained. The probabilistic alternation between TVG and Invert-TVG tasks is intuitive and well justified experimentally.
3. Three inversion tasks. The decomposition into VC, AR, and VD tasks provides a balanced approach across understanding levels, which are aligned with the grounding objective.
4. Quantitative improvements. Results on Charades-STA, ActivityNet, and QvHighlight show consistent gains.
5. Good analysis and ablations. The ablation study provides insight into the contribution of each inversion task and the effect of task probability, which helps validate the design decisions.

**Weaknesses:**

1. Teaser figure clarity (Fig. 1). The sequence of frames is visually unclear, and it is difficult to understand the action being performed. Consider refining the teaser fig.
2. Limited per-component ablation. Table 2 investigates single-task vs. multi-task setups but does not test pairwise combinations (e.g., VC+AR, AR+VD, VC+VD). Adding these 3 additional experiments would more clearly quantify each component’s interaction.
3. Missing recent related work. The paper misses recent temporal grounding works such as Number it: Temporal Grounding Videos like Flipping Manga (CVPR 2025) and TimeSuite (ICLR 2025), both relevant to grounding and MLLM temporal reasoning. These should be discussed for completeness.
4. Limited qualitative diversity. The qualitative examples (Figs. 7–9) mostly feature human actions like “opening/closing doors.” More varied examples (e.g., multi-person or non-human object interactions) would strengthen the claim of generalizable action understanding.

**Questions:**

see above

---

> ### Author Response · Authors · 2025-11-21
>
> **Q1**: Teaser figure clarity (Fig. 1).
>
> **Answer**: Thanks very much for the kind comment. We will revise the teaser figure as suggested. Please refer to the updated paper later. Thanks a lot!
>
> **Q2**: Add pairwise ablations: VC+AR, AR+VD, VC+VD.
>
> **Answer**: We sincerely thank the reviewer for this helpful suggestion. As recommended, we have conducted additional ablation studies on the pairwise component combinations, as shown in the following updated table.
>
> |  Method     | R1@0.3 | R1@0.5 | R1@0.7 |
> | ---------- | ------ | ------ | ------ |
> | Only-TVG   | 78.7   | 64.1   | 36.9   |
> | Only-VD    | 79.1   | 64.3   | 39.4   |
> | Only-AR    | 78.2   | 65.2   | 43.8   |
> | Only-VC    | 78.8   | 68.0   | 42.0   |
> | AR+VD      | 79.6   | 67.9   | 43.6   |
> | VC+AR      | 78.8   | 68.1   | 43.8   |
> | VC+VD      | 80.0   | 68.5   | 42.1   |
> | Invert4TVG | 80.8   | 69.0   | 44.0   |
>
> As can be seen, the combination of VC and AR improves R1@0.7 to 43.8, outperforming either task alone (VC: 42.0; AR: 43.8), indicating complementary benefits between verb completion and action recognition. The VC+VD pair achieves the highest R1@0.3 (80.0) among two-task setups, suggesting that video description aids verb-focused localization. Invert4TVG (integrating all three auxiliary tasks) achieves the best overall results (R1@0.7: 44.0), demonstrating that multi-task synergy is maximized when all components are jointly optimized.We will revise Section 4.3 to incorporate these analyses.
>
> **Q3**: Missing recent related work Number It (CVPR25) and TimeSuite (ICLR25).
>
> **Answer**: Thanks a lot for the comments. TimeSuite was already discussed and cited in our paper, as seen in Table 1.
>
> We will cite Number It. The following is the comparison between Number It and our method, with the results of Number It from their paper. As can be seen, our method outperforms the approach on the Charades-STA dataset.
>
> | Model                        | R1@0.3 | R1@0.5 | R1@0.7 |
> |------------------------------|--------|--------|--------|
> | Qwen2-VL-7B+NumPro(7B)       | 60.7   | 36.8   | 15.9   |
> | LongVA-7B-DPO+NumPro-FT(7B)  | 63.8   | 42.0   | 20.6   |
> | Invert4TVG(7B)               | 83.0   | 72.5   | 51.4   |
>
> **Q4**: Limited diversity of qualitative examples.
>
> **Answer**: Thanks a lot for the nice suggestion. We have selected more diverse qualitative results, including multiple persons, non-human object interactions, multiple actions by one person, actions with temporal sequence relationships, actions that involve causal reasoning, etc. We will add these results into the supplemental material.  Please refer to the supplemental material later for more information.

---

> > ### Comment · Reviewer_9Nkj · 2025-11-25
> >
> > I thank the authors for their response. My concerns are addressed, I’m willing to increase my score.

---

> > > ### Author Response · Authors · 2025-11-26
> > >
> > > The authors sincerely thank the reviewer's feedback.

---

### Official Review · Reviewer_voLD · 2025-10-30

**Soundness:** 3
**Presentation:** 3
**Contribution:** 2
**Rating:** 6
**Confidence:** 4

**Summary:**

This paper introduces Invert4TVG, a framework for Temporal Video Grounding (TVG). The key motivation is that current TVG models (e.g., Time-R1) tend to over-optimize temporal IoU while neglecting action understanding, leading to semantically incorrect grounding despite high IoU scores. The authors propose three inversion-based auxiliary tasks (Invert-TVG) derived from the original TVG supervision. Experiments on some popular datasets show SOAT performance.

**Strengths:**

+ The idea of reversing the TVG process to construct self-supervised auxiliary objectives is conceptually fresh and well-motivated.
+ The method is compatible with large LVLMs and scalable to different model sizes (3B and 7B).
+ The article is written in a prominent style, making it easy for readers to grasp the core points.

**Weaknesses:**

+ The authors claim that “existing TVGs over-optimize IoU, leading to semantic degradation”, but this paper lacks the experimental analysis of IoU improvement.
+ The datasets used in TVG and Invert-TVG duplicates? Is the Invert-TVG more like a video QA task?
+ Limited performance improvement compared to Time-R1. There are no related ablation experiments for parameter p=0.8 in the main paper.

**Questions:**

Please refer to Weaknesses.

---

> ### Author Response · Authors · 2025-11-21
>
> **Q1**: Lack Experimental Analysis of IoU Improvement.
>
> **Answer**: In our paper, **Table 1** presents the IoU results of the compared and our method. Thanks for reminding us that the analysis of this data might be insufficient. We will revise the paper to add more thorough analysis. For example, the paragraph of "Comparisons on the Charades-STA dataset with Fine-Tuning" will be revised to:
> >**Comparisons on the Charades-STA dataset with Fine-Tuning.** As detailed in Table 1, our method, Invert4TVG, demonstrates superior performance across all IoU thresholds (R1@0.3, R1@0.5, R1@0.7), providing concrete evidence against the semantic degradation caused by IoU over-optimization in existing methods. Specifically, our 7B variant achieves state-of-the-art results, with an R1@0.7 of 51.4, which not only surpasses the strongest fine-tuned baseline, Time-R1* (7B) at 50.1, but also significantly outperforms other types of models like TimeSuite (43.0) and SnAG (VLP, 46.2). More importantly, the more pronounced improvements are observed at the higher IoU threshold (R1@0.7) for both our 3B and 7B models, which strongly support our claim. For instance, our 3B model achieves a notable R1@0.7 of 44.0, a substantial lead over the 3B Time-R1 baseline (36.9). This IoU gain, especially under stricter localization criteria, validates that our approach effectively leads to IoU improvement through better action comprehension.
>
> Later, please refer to the updated PDF for more information.
>
> **Q2**: Datasets used in TVG and Invert-TVG duplicates?
>
> **Answer**: Yes, the training data for Invert-TVG is entirely modified from the TVG training data by swapping the inputs and outputs of the TVG data.
>
> **Q3**: Is the Invert-TVG a video QA task?
>
> **Answer**: Invert-TVG is **not** a video QA task. Given a video (i.e., the cropped Ground-Truth video segment corresponding to a user query), InvertTVG generates description of the video, extracts verbs from the description, and finally compares these verbs with the ground-truth counterparts in the query. Therefore, it is very different from VQA.
>
> **Q4**: Limited performance improvement compared to Time-R1.
>
> **Answer**: For Time-R1, we compared two versions: Time-R1 and Time-R1*. Time-R1 was trained on the same dataset as our method, while Time-R1* was trained on multiple additional datasets (including YT-Temporal, DiDeMo, QuerYD, InternVid, and HowTo100M).
>
> Compared to Time-R1, our approach shows substantial gains. For example, at R1@0.3, our 3B model achieves 80.8 versus Time-R1 (3B)'s 74.6, and our 7B model reaches 83.0 versus Time-R1 (7B)'s 78.1.
>
> Even when compared to Time-R1*, our method is highly competitive. For example, our 3B model scores 80.8 versus 78.7, and our 7B model achieves 83.0 versus 82.8.
>
> **Q5**: No ablation experiments for parameter p=0.8 in the main paper.
>
> **Answer**: We would like to clarify that Figure 5 in the main text provides a comprehensive ablation analysis of the hyperparameter $p$. As illustrated in the figure, we systematically evaluated $(1-p)$ across a range of values: 0, 0.2, 0.4, 0.6, 0.8, and 1.0. The results show that setting $p=0.8$ yields the best performance.
>
> We will make this explanation more explicit in the revised manuscript to avoid any confusion.

---

### Official Review · Reviewer_nj73 · 2025-10-31

**Soundness:** 3
**Presentation:** 3
**Contribution:** 3
**Rating:** 6
**Confidence:** 3

**Summary:**

This paper introduces a novel approach to address the common issue of insufficient action understanding in Temporal Video Grounding (TVG) models, often stemming from an over-optimization of Intersection-over-Union (IoU). The core contributions of the paper include identifying the degradation of action understanding due to IoU over-optimization, and subsequently proposing three self-supervised inversion TVG tasks—Verb Completion, Action Recognition, and Video Description—derived from original TVG annotations to enhance the model's semantic understanding of actions across different granularities. The framework employs a dynamically balanced reinforcement learning strategy, executing the primary TVG task with high probability and interleaving it with lower-probability inversion TVG tasks. This ensures the model maintains robust temporal localization while continuously reinforcing its action understanding. Invert4TVG consistently outperforms state-of-the-art approaches across various benchmarks, showcasing its superior capability in understanding complex actions.

**Strengths:**

1.The paper introduces the unique "Inversion TVG Tasks" combined with a dynamically probabilistic reinforcement learning framework, cleverly leveraging existing data for self-supervised action understanding, providing a novel problem-solving and execution strategy for the TVG field.
2.The methodology is rigorously designed, especially in the three multi-granularity inversion tasks and their reward functions. Experiments are comprehensive, yielding significantly superior SOTA performance across multiple benchmarks (including zero-shot settings), further validated by thorough ablation studies, making the conclusions robust.
3.The paper is logically structured and lucidly written. High-quality visualizations are used effectively to make complex motivations and the proposed framework highly intuitive and easy to comprehend.
4.This work not only addresses a key bottleneck in the TVG field, significantly advancing the state of the art, but its core idea of "inversion tasks" also holds broad inspirational value, offering a new paradigm for other multimodal alignment and understanding tasks.

**Weaknesses:**

1.The paper’s central claim is that "by reversing the task, the model’s action-understanding ability is preserved and even enhanced," and this improvement is presented as the reason for the superior Temporal Video Grounding (TVG) performance. However, throughout the experimental section the authors only report higher TVG localization metrics (R1@m). They never directly evaluate the final Invert4TVG model on the action-understanding tasks (i.e., the three reversed tasks: VC, AR, and VD) on the test set. Consequently, one can only infer—indirectly—from the improved localization accuracy that action understanding has become stronger, which lacks direct evidence. The authors should supplement the corresponding experiments to provide conclusive proof for the core argument that "action-understanding ability is improved."

2.The notion of "action understanding" is overly simplified into superficial verb matching, and its binary reward mechanism cannot distinguish genuine semantic comprehension from opportunistic keyword generation. Such a narrow proxy disregards the deep understanding of context, entity properties, and logical relations indispensable for the TVG task. Consequently, any improvement may reflect only enhanced pattern-matching rather than true generalizable understanding, fundamentally undermining the scientific explanatory power of the core claim—"boosting localization performance by enhancing comprehension."

3.A fundamental limitation of this methodology is its atomistic, verb-centric design, which treats isolated verbs as the basic unit and inherently disregards the compositional nature of language. Consequently, it cannot handle complex queries that involve attributes, temporal order, or logical relations—examples include “event B happens after event A” or “the action performed by the person wearing red.” The method’s success is therefore likely confined to simple, atomic queries and fails to generalize to the richer, more complex TVG scenarios common in real-world applications. The reported inability to understand “again” is not an incidental error but a direct symptom of the approach’s intrinsic deficiency in modeling relational information, which severely restricts the universality and practical value of its contributions.

**Questions:**

1.The core hypothesis of the paper is that reversing the task (Invert-TVG tasks) enhances the model’s action-understanding ability, which in turn improves temporal video grounding (TVG) performance. At present, however, this is only an indirect inference drawn from the final localization results. How do you plan to verify that the model’s action-understanding ability has indeed been improved, thereby boosting its TVG performance?

2.The verb-centric design appears to fundamentally limit the model’s ability to handle complex language. Your own failure case (failing to understand “again”) already hints at this, and it does not seem to be a minor issue. Could you discuss more deeply how your method performs and where it breaks down when faced with compositional queries that contain:
1. entity attributes (e.g., “the person wearing red”);
2. strict temporal relations (e.g., “after A, before B”);
3. logical or causal relations (e.g., “tried to do something but failed”)?

3.Your simplistic binary reward—full credit for matching only the verb—invites reward hacking: the model can earn a perfect score by producing meaningless sentences that happen to contain the target verb, instead of truly understanding the video. How can knowledge acquired through such "short-cutting" effectively benefit the main TVG task, which requires deep comprehension? Moreover, why did you opt for this minimal reward rather than metrics that better evaluate content quality (e.g., semantic similarity) to mitigate this risk?

---

> ### Author Response · Authors · 2025-11-21
>
> **Q1**: How to verify model's Action-Understanding Ability?
>
> **Answer**: Thank you for this important question, and the useful suggestion.
>
> To assess whether the proposed model improves the action-understanding capability, a direct way is to evaluate the model on the three auxiliary tasks (VC, AR, and VD). The results are shown in the following tables:
>
> Charades Dataset
> | Method        | TVG mIoU | VC Accuracy | AR Accuracy | VD Accuracy |
> | ------------- | -------- | ----------- | ----------- | ----------- |
> | Qwen2.5-VL-3B | 24.9  | 0.288  | 0.307       | 0.433 |
> | Time-R1       | 56.0  | 0.265 | 0.277  | 0.286   |
> | Invert4TVG    | 58.0 | 0.579 | 0.656  | 0.421   |
>
> Qvhighlight Dataset
> | Method        | TVG mIoU | VC Accuracy | AR Accuracy | VD Accuracy |
> | ------------- | -------- | ----------- | ----------- | ----------- |
> | Qwen2.5-VL-3B | 10.700| 0.201| 0.318 | 0.407|
> | Time-R1       | 21.400| 0.184 | 0.225| 0.210|
> | Invert4TVG    | 23.000| 0.320| 0.519| 0.412|
>
> As shown, after training with only IoU rewards (Time-R1), the model’s performance on VC, AR, and VD drops compared to the base model (Qwen2.5-VL-3B). In contrast, Invert4TVG enhances the performance of VC, VD, and VR, while also improving the mIoU.
>
> **Q2**: Verb-centric design ≠ genuine semantic comprehension. How to handle complex queries that involve attributes, temporal order, or logical relations?
>
> **Answer**: We sincerely thank the reviewer for this very insightful comment.
>
> The verb-centric tasks we designed are simple proxy tasks. When trained extensively, they facilitate the model's ability to achieve text/video semantic understanding. Such proxy tasks are common used in many fields. For instance, LLM training often involves simply predicting the next word, and BERT uses word filling.
>
> The reason motivating us to design verb-centric tasks is that: our analysis of the Qwen2.5-VL-3B results revealed that a significant portion of its errors stemmed from a fundamental misunderstanding of actions. Regarding the reviewer's point about complex queries involving attributes, temporal order, or logical relations, we have the following views:
>
> - For entity attributes (e.g., "the person wearing red"), we found that recognizing them is a relative strength of modern foundation models. For instance, identifying "a person wearing red" or "a man in a black shirt" is typically an easy task for large models nowadays.
>
> - For many cases with strict temporal relations and logical/causal relations, they often play a secondary, albeit important, role in video grounding. For example:
>    - In "He tried to open the door but failed", while "failed" indicates negation, the core action that needs to be understood and localized is "open the door".
>    - In "She ran because she was late", the causal clause "because she was late" provides context, but the key action is "ran".
>    - In "After closing the window, he sat down", the ordinal/temporal cue "after" defines the sequence, but successful localization depends on correctly identifying the two core actions: "closing" and "sat down".
>
> We collected 1000 samples for "Entity Attributes", 600 for "Temporal Relations", and 500 for "Logical/Causal Relations". We conducted comparative experiments on these samples:
>
>
> | Relation Type   | Model | mIoU | R1@0.3 | R1@0.5 | R1@0.7 |
> | ------------------------------------------------------------ | ------------- | ---- | ------ | ------ | ------ |
> | **Entity Attributes** (e.g.  "Man in black shirt works around his grill", "Person wearing red walks" ). 1000 samples in total*. | Qwen2.5-VL-3B | 5.0  | 6.8    | 3.2    | 1.6    |
> |     | Invert4TVG    | 24.4 | 33.7   | 21.8   | 10.5   |
> | **Strict Temporal Relations** （e.g., "After opening the door, he walks inside", "While cooking, she watches TV"). 600 samples in total. | Qwen2.5-VL-3B | 5.3  | 10.5   | 2.1    | 0.5    |
> |                                                              | Invert4TVG    | 22.9 | 34.2   | 23.6   | 7.8    |
> | **Logical/Causal Relations** (e.g., "He tried to open the door but failed", "Despite being tired, she continued working"). 500 samples in total. | Qwen2.5-VL-3B | 13.8 | 16.6   | 14.2   | 9.5    |
> |                                                              | Invert4TVG    | 34.9 | 71.4   | 47.6   | 23.8   |
>
> *The samples are selected from Charades-STA, QVHighlights, and ActivityNet by an LLM given few-shot demonstrations.
>
> As shown by the experimental results above, our method performs much better on these complex examples compared to Qwen2.5-VL.
>
> However, our method does sometimes make errors with certain adverbs that carry specific semantic meanings, such as "again". Similar adverbs include "once more", "repeatedly", etc. A potential reason is that the model might overlook these words. To address this, one solution is to design similar tasks for these key adverbs (e.g., "again") as we did for verbs in the VC, VD, and AR tasks.

---

> > ### Author Response · Authors · 2025-11-21
> >
> > **Q3**: Reward hacking problem.
> >
> > **Answer**: Thank you for this insightful comment.
> >
> > First, we would like to point out that the VC and AR tasks do not suffer from the Reward Hacking problem.
> >
> > For VD, which generates a description of a video while requiring that the description includes the action mentioned in the query, experiments have shown that VD generally does not produce meaningless video descriptions. This is because these descriptions are generated based on the actual video content, rather than being arbitrary. For example:
> > - **Video**: CUQYX.mp4
> >    - **Query**: "Person **throws** the bag against the wall softly."
> >    - **VD Generated Description**: "A man is sitting on the floor in a hallway, holding a plastic bag. He is wearing a plaid shirt and black pants. He looks at the camera and then **throws** the bag. He then sits back down and looks at the camera again."
> >
> > - **Video**: 6JLD4.mp4
> >    - **Query**: "A person **turns on** a light in the closet."
> >    - **VD Generated Description**: "A man is standing in front of an open closet door, looking inside. He is wearing a gray shirt and black pants. He is holding a white object in his hand. He is looking at the clothes hanging in the closet. He **turns** on a light in the closet."
> >
> > As these examples show, although the generated video descriptions do not perfectly match the queries, the described video content itself is realistic and plausible. The requirement for these descriptions to include the action from the query aims to encourage the model to detect and identify these actions within the video, thereby enhancing the model's perception and understanding of actions.

---

> > > ### Comment · Reviewer_nj73 · 2025-11-27
> > >
> > > Thanks for the authors' responses. I think my concerns have been addressed.

---

> > > > ### Author Response · Authors · 2025-11-28
> > > >
> > > > Thanks very much!

---

### Official Review · Reviewer_tE1U · 2025-11-01

**Soundness:** 3
**Presentation:** 3
**Contribution:** 3
**Rating:** 6
**Confidence:** 3

**Summary:**

This paper introduces Invert4TVG, a GRPO-based reinforcement learning framework for temporal video grounding that mitigates the loss of action understanding caused by optimizing solely for IoU. The core idea is to invert the TVG mapping via three auxiliary tasks—Verb Completion, Action Recognition, and Video Description—and interleave them with the main TVG objective through probabilistic scheduling (≈80% TVG, 20% Invert-TVG). On Charades-STA, Invert4TVG improves R1@0.7 by 7.1 points over Time-R1, demonstrating the benefit of coupling localization with semantics.

**Strengths:**

1) **Clear diagnosis and motivation.**
   The paper convincingly shows that **IoU-centric optimization can erode action understanding**, with Figure 1 and accompanying evidence making the point concrete. The “buttoning vs. unbuttoning” example effectively illustrates the failure mode.

2) **Strong empirical gains.**
   The method reaches **44.0% R1@0.7 on Charades-STA**—a **+7.1** point improvement over Time-R1—and delivers **consistent improvements across Charades-STA, ActivityNet, and QVHighlights**. Notably, both the **3B** and **7B** variants benefit.

3) **Practical auxiliary task design.**
   The three **inversion tasks** (Verb Completion, Action Recognition, Video Description) **reuse existing TVG annotations** and require **no extra data collection**, making the approach pragmatic and easy to adopt.

**Weaknesses:**

1) **Incremental novelty via training reformulation.**
   The main ingredients—GRPO-based RL, inversion-style auxiliary tasks, and template/format rewards—are adapted from existing ideas. The contribution lies primarily in **recasting TVG training** rather than introducing a fundamentally new algorithmic primitive.

2) **Verb-centric semantics may be brittle.**
   Reliance on **SpaCy verb lemmatization** for VC/AR/VD risks overlooking **non-verbal cues** (objects, states) and **nuanced modifiers** (e.g., adverbs, negation, ordinality), potentially limiting semantic coverage.

3) **Unclear computational overhead.**
   The paper does not quantify the **runtime/VRAM costs** introduced by the three auxiliary tasks (data processing, reward computation, extra forward/backward passes). A concise **cost table** and throughput metrics would clarify practicality.

4) **Narrow baseline scope.**
   Comparisons focus largely on **Time-R1** (concurrent work). Including **multi-task/regularization** baselines (e.g., auxiliary captioning, CLIP-style consistency, feature-level regularizers) would better contextualize the benefits of the inversion tasks.

**Questions:**

Please refer to the Weaknesses

---

> ### Author Response · Authors · 2025-11-21
>
> **Q1**: Recasting TVG training is novel, but no new algorithmic primitive.
>
> **Answer**: Thank you for the valuable comments. As pointed out, one of our contributions is "recasting TVG training" into a framework that simultaneously trains both TVG and Invert-TVG tasks, aiming to address the limitations of relying solely on mIoU-based losses.
>
> From the algorithm perspective, our work presents the following contribution: within the GRPO-based RL framework, we propose a probability-based multi-task optimization method. The purpose is to effectively resolve the joint optimization challenges arising from **tightly coupled** tasks where inputs and outputs are exchanged between the tasks.
>
> **Q2**: Verb-centric semantics may be brittle.
>
> **Answer**: Thank you for this insightful comments.
>
> We adopted the verb-centric designs because of the following reasons. Our analysis of the Qwen2.5-VL-3B results revealed that a significant portion of its errors stemmed from a fundamental misunderstanding of actions. This motivated us to design a method that preserves the model's action comprehension capability.
>
> But we thank very much for reminding us “objects/states” and “adverbs, negation, ordinality” in the query.
> - For "objects/stages", we found modern foundation models to be relatively strong at recognizing them. For example, identifying a "person" or a "desk" is an easy task for current large models.
> - For “adverbs, negation, and ordinality”, we found that the modifiers often play a secondary, albeit important, role. For example (the following examples of queries are from the Charades and Qvhighlight datasets):
>    - In "He tried to open the door but failed", while "failed" indicates negation, the core action to be understood and localized is still "open the door".
>    - In "She ran because she was late", the causal clause "because she was late" provides context, but the key action is "ran".
>    - In "After closing the window, he sat down", the ordinal/temporal cue "after" defines the sequence, but successful localization depends on correctly identifying the two core actions, "closing" and "sat down".
>
> We collect 1100 the above kinds of examples from Charades, QvHighlight, and ActivityNet datasets, and compare Invert4TVG with Qwen2.5-VL on these complex and challenge examples. The results are as follows:
>
> - Examples with strict temporal relations (e.g., "after A, before B") (600 samples* in total)
> | Model             | mIoU | R1@0.3 | R1@0.5 | R1@0.7 |
> | ----------------- | ---- | ------ | ------ | ------ |
> | Qwen2.5-VL-3B     | 5.3  | 10.5   | 2.1    | 0.5    |
> | Invert4TVG (Ours) | 22.9 | 34.2   | 23.6   | 7.8    |
> >*We use LLM to automatically select the kind of complex queries given few-shot demonstrations.
> - Examples of logical or causal relations (e.g., "tried to do something but failed") (500 samples)
> | Model             | mIoU | R1@0.3 | R1@0.5 | R1@0.7 |
> | ----------------- | ---- | ------ | ------ | ------ |
> | Qwen2.5-VL-3B     | 13.8 | 16.6   | 14.2   | 9.5    |
> | Invert4TVG (Ours) | 34.9 | 71.4   | 47.6   | 23.8   |
>
> The above results show that our method demonstrates strong generalization ability on complex queries with adverbs, negation, and ordinality.
>
> **Q3**: Unclear computational overhead.
>
> **Answer**: We sincerely thank the reviewer for this helpful comment. Below, we provide a detailed breakdown of the computational overhead introduced by the three auxiliary tasks (data processing, reward computation, and extra forward/backward passes), including concise cost tables.
>
> The following table shows separate runtime costs per task.
>
> | TVG  | VD   | VC   | AR   | Total |
> | ---- | ---- | ---- | ---- | ----- |
> | 65h  | 6.5h | 5h   | 3.5h | 80h   |
>
> As can be seen from the table, in our method, TVG overall consumed 65 hours (accounting for 81.25%), while Invert-TVG overall consumed 15 hours (accounting for 18.75%), with VD, VC, and AR consuming 6.5 hours, 5 hours, and 3.5 hours, respectively. It is important to note that if the Invert-TVG tasks were not used and only TVG tasks were employed, the total training time would still amount to 80 hours. The Invert-TVG task does not lead to an increase in training time, and our method just took the training time originally meant for the TVG task and used it for the Invert-TVG task.
>
> Detailed time usage in the three auxiliary tasks per training step is shown in the following table:
>
> |      | Data Processsing | Forward | Reward | Backward |
> | ---- | ---------------- | ------- | ------------------ | -------- |
> | VC   | Offline* | 117ms   | 81ms   | 89ms     |
> | VD   | Offline | 121ms   | 46ms  | 90ms     |
> | AR   | Offline  | 119ms   | 10ms  | 90ms     |
> >*The data processing was fulfilled before the training.
>
> Similarly, the VRAM costs are nearly identical to the baseline, with no additional overhead.
> | Model          | VRAM costs |
> | -------------- | ------------------ |
> | Qwen-vl-2.5-3B | 146G|
> | Our | 142G |

---

> > ### Author Response · Authors · 2025-11-21
> >
> > **Q4**: Narrow baseline scope.
> >
> > **Answer**: Thanks very much for the comments. As suggested, we have conducted additional experiments exploring alternative auxiliary tasks, instead of our inversion-based tasks.
> >
> > The core of our method is to take the cropped Ground-Truth video segment as input and output query-based content. For comparison, we designed different auxiliary tasks. The key is to use the entire video as input, rather than the cropped GT video segments.
> >
> > - Experiment 1: Auxiliary Captioning. In this experiment, we used the full video as input, while using the concatenation of all ground-truth queries (a video may have multiple query annotations) as the target.
> > - Experiment 2: CLIP-Extracted Features and Regularization. We used the full video as input, and output a video segment. Then, we extracted features from the video segment and the query by CLIP, and computed similarity between the two kinds of features. Finally, we used the similarity as the reward loss.
> >
> > Due to time constraints, we trained the variant models for 1 epoch. Our experiments were conducted on the Charades-STA dataset, as this allowed us to reuse the checkpoint model saved after 1 epoch of training using our method.
> >
> > As shown in the following table,  the alternative approaches did not yield satisfactory results. For the first experiment, generating description for the entire video encourages the model to capture a broad set of actions and events, which prevents the fine-grained understanding of the specific action required for temporal localization. In the second experiment, while the approach encourages the alignment between the text and the video, clip-based similarity is not entirely reliable, which can produce a high score for a segment and a query that are actually not a good match.
> >
> > | Model                                      | mIoU | R1@0.3 | R1@0.5 | R1@0.7 |
> > | ------------------------------------------ | ---- | ------ | ------ | ------ |
> > | Auxiliary Captioning                       | 49.2 | 77.5   | 57.5   | 32.5   |
> > | CLIP-Extracted Features and Regularization | 48.5 | 71.0   | 56.0   | 34.5   |
> > | Invert4TVG (Ours)                          | 53.4 | 75.5   | 65.0   | 35.5   |

---

### Author Response · Authors · 2025-12-03
**To Area Chair and Reviewers**

We sincerely thank all the reviewers for their insightful feedback and valuable time. All reviewers have recommended acceptance with helpful comments. During the rebuttal period, we have thoroughly addressed the raised questions through extensive experiments and clarifications. After reading the rebuttal, Reviewers nj73 and 9Nkj confirmed that their concerns have been satisfactorily addressed. In particular, Reviewer 9Nkj raised the score from 6 to 8. Due to the system disclosure issue, Reviewers tE1U and voLD were unfortunately unable to provide their feedback.

In this paper, we propose a reinforcement learning framework (Invert4TVG) for Temporal Video Grounding (TVG) that addresses the problem of degraded action understanding caused by over-optimizing for temporal IoU. As endorsed by the reviewers, the core innovation is the idea that repurposes TVG annotations into inverted ones to enhance semantic comprehension at various granularities. Besides, we also propose a probability-based training strategy that unifies the training of the multiple conflict tasks. The shown SOTA results under various benchmarks prove the effectiveness of our model. This novel work with good results will definitely create a very positive impact to the video understanding community.

We sincerely hope that this SOTA paper on an important topic will be considered suitable for publishing in ICLR 2026. Thank you for considering our paper for the conference.

---

### Meta-Review · Area_Chair_YSYW · 2026-01-10

**Summary:**

All reviewers were positive about this paper, appreciating the well-motivated and interesting idea and the comprehensive experiments. The main concerns were about incremental novelty, missing analyses/baselines, and unclear writing. The authors’ rebuttal addressed most of the concerns, clarifying missing details and providing additional experimental results. After discussion, reviewers nj73 and 9Nkj confirmed that their concerns have been addressed. AC also finds that the idea of reversing the TVG process to construct self-supervised auxiliary objectives interesting, and the authors’ response resolves most of the critical concerns raised by the reviewers, thus recommending acceptance.

**Reviewer Concerns:**

The rebuttal clarified the issue of limited novelty, noting their technical contribution of developing a probability-based multi-task optimization method within the GRPO-based RL framework, and also provided missing details and additional experiments to address the issues of missing analyses and unclear writing. AC finds no outstanding critical issues.

**Reviewer Scores:**

Reviewer tE1U  would have changed the score from 6 to 8.
Reviewer nj73 retained the original score 6 after discussion.
Reviewer voLD  would have changed the score from 6 to 8.
Reviewer 9Nkj  retained the original rating of 8 after discussion.

---

### Decision · Program_Chairs · 2026-01-26

Accept (Poster)